# "Why Here?"—Pull Factors for the Attraction of Non-EU Immigrants to Rural Areas and Smaller Cities

Elisete Diogo [1,2]

1 CARE-Research Center on Health and Social Sciences, VALORIZA—Centro de Investigação para a Valorização de Recursos Endógenos, Instituto Politécnico de Portalegre, 7300-110 Portalegre, Portugal; elisetediogo@gmail.com
2 Católica Research Centre for Psychological, Family and Social Wellbeing (CRC-W), Universidade Católica Portuguesa, 1649-023 Lisbon, Portugal

**Abstract:** The 2030 Agenda for Sustainable Development recognizes the crucial role of the regional dimension for economic, social, and environmental development. Sustainable development may be linked to migration management to strategically disperse international migrants to regions in need of ameliorating rural challenges. This paper explores the features affecting international migrants' intentions to move to rural areas, such as Alentejo, Portugal, based on a set of micro-, mezzo-, and macro-sociological migration theories to support policymakers. This paper addresses the following research question: what motivates immigrants' decisions to move to rural regions, such as Alentejo, Portugal? Practitioners (*n* = 8) and migrants (*n* = 15) were interviewed, and then a thematic analysis supported by MaxQDA 2022 was conducted. The results suggest that there is a set of motives for international migrants to move to rural areas and smaller cities based on multilevel factors, both economic and non-economic, such as the following: employment availability and promises of work; lower living costs compared to bigger cities; quality of life; local services support; and echoes of the country of origin. Migrants' networks and seeking greater opportunities were consistent motives. The pull to rural areas, however, is a side effect of the attraction of Portugal and Europe as destinations. The conclusions highlight implications for policy and practice on migration and local development.

**Keywords:** international migration; depopulated regions; motives to go to rural areas; immigrants in rural regions; regional development

## 1. Introduction

The 2030 Agenda for Sustainable Development recognizes the crucial role of the regional dimension for economic, social, and environmental development and to promote inclusive societies during sustainable development (United Nations 2015). The aims of sustainable development are to create effective responses—with a cross-disciplinary perspective—to complex challenges and issues, including migration and ecosystem resilience (Suciu and Năsulea 2019). This demands a multilevel and multilateral approach, but there is room in the academic agenda to give more importance to the topic of sustainability and sustainable development (Secundo et al. 2020). The present paper is thus relevant for focusing on the inclusion and sustainability of rural areas, such as the Portuguese Alentejo region, which are characterized as depopulated areas and lack economic competitiveness and innovation (Gauci 2020; Mota 2019). Interviews gave a voice to practitioners from non-governmental organizations (NGOs) and from municipalities and international immigrants, with a sample of 23 participants. Despite being a small sample, the theoretical saturation criterium (Flick 2005, 2013) was guaranteed. Given the insufficient research on migration to rural places, this study is innovative, particularly for this region, and because it provides specific motives for moving there. This paper aims to inform policymakers of the value of international migrants, who can then design evidence-based plans to attract

international migrants to support sustainable local development (Butkus et al. 2018; Diogo 2024; Diogo et al. 2023; Gauci 2020; Quintino 2018).

The aspirations for the EU's rural areas in 2040 are captured in ten shared goals, including inclusive communities of inter-generational solidarity, fairness, and renewal, which are open to newcomers and which foster equal opportunities for everybody. Furthermore, they include building attractive spaces, places of diversity, dynamic communities focusing on wellbeing, as well as engagement in place-based and multi-level governance (European Union 2023). This point is shared by Gauci (2020). These aspirations are aligned with the view that international migrants[1] are demographic and socio-economic resources for the host society (Diogo 2024; Góis 2023; Oliveira 2022). Although there are several challenges for sustainability, there is an increasing reliance on migrant labor as a means of sustaining businesses that would otherwise be unviable (McAreavey and Argent 2018). The current study area, namely the Alentejo region, is characterized by rural areas and smaller cities and can thus be framed as a low-density territory (Deliberação no. 55/2015), which indicates depopulation; poor access to public services and employment; low accessibility; lack of economic competitiveness and innovation; as well as precarious governance (Carvalho and Oliveira 2017; ESPON 2018; Morén-Alegret et al. 2021; Mota 2019).

The international literature also suggests that the issues of smaller cities and rural areas, including depopulation and the lack of workforce, can be addressed through international migration. International migration can also guarantee the continuity of primary services (Butkus et al. 2018; Diogo 2024; Diogo et al. 2023; Gauci 2020; Quintino 2018). The results of Collantes et al. (2013) show that the arrival of immigrants has contributed substantially to reducing and even halting or reversing depopulation. International immigrants are adding value and playing a role in the sustainability and development of rural areas through, for example, international migrant entrepreneurship (Morén-Alegret et al. 2021), the richness of cultural and religious diversity, and social security contributions (Diogo et al. 2023; Gauci 2020; Laine et al. 2023).

Initiatives to contribute to attracting immigrants could thus support the sustainable development of these regions (Diogo 2024; Sampedro and Camarero 2018), and, in fact, they have increasingly begun to attract third-country nationals as immigrants (from a European Union perspective) in considerable numbers (Collantes et al. 2013). McAreavey (2017) acknowledges that new immigration destinations are emerging across the globe as international migrants arrive in rural regions and small towns, many of which have a limited history of immigration, for example, localities in non-metropolitan Australia, Northern Ireland, and America. An effective policy plan to attract immigrants must be supported by evidence (European Commission 2023b) about the pull factors for an individual or a family to move to another country. This topic has also been extensively studied in the sociology of migration (e.g., Góis 2023; Peixoto 2004). However, because of the complex and changing nature of this phenomenon, certain features on migration remain largely unexplored (De Haas 2021; McAreavey 2017). Pull factors for migration to disadvantaged rural regions and smaller cities, such as the Alentejo region of Portugal, have seldom received attention from scholars.

## 1.1. Micro- and Macro-Level Theories Explaining Migrations

Migration, including both internal and international flows, has been studied in different fields—including economics, geography, demography, political science, and sociology—since the 19th and 20th centuries. Understanding this phenomenon is important given its impacts, including disrupting general living standards, culture shock and acculturation, new contacts, and ethnic conflicts, as well as burdens on infrastructure, such as public transport systems and education (Peixoto 2004).

To explain the existence of migration, sociologists have proposed two wide groups of theories: (a) micro-sociological causes, such as an individual's choice, and (b) macro-sociological causes, which are a result of social strengths. A third group, mezzo-sociological

causes, has also been identified, which focuses on family ties, social networks, peer groups, and community bonds (Migration Data Portal 2021; Wadood et al. 2021).

The push–pull model (see Lee 1966), which is one of the micro-level theories of international migration, explains migration as being rooted in an individual's economic will. One decides to migrate based on information from the country of origin and the potential destination country with the aim of improving one's economic conditions. Wadood et al. (2021) clarifies Lee's thesis:

> ""push" factors operate in the economically backward regions or countries of the world (insufficient demand of labour and low wages, forcing people to search for alternative locations for better livelihoods), whereas "pull" factors operate in the economically advanced regions or countries of the world (higher demand for labour and higher wages, encouraging people to come in and stay there)." (p. 54)

Economic aspects have a wide range and go beyond employment. Peixoto (2004) notes that scholars admit enlarging the model and including non-economic aspects such as changing value systems or political and religious contexts. There are also intervening variables such as distance and travel costs between countries, household characteristics, support networks, migratory policy, and cultural and linguistic identity. There are also additional personal factors, such as life cycle position, contacts, or information sources.

De Haas (2021), observing the contemporary world, noted "thinking on migration remains implicitly or explicitly based on simplistic push–pull models or neo-classical individual income [...] despite their manifest inability to explain real-world patterns and processes" (p. 1). This argument leads us to explore other complementary theories to explain migrations from various analytical perspectives.

Another micro-level theory is human capital theory (see, e.g., Becker 1964), which includes similar factors but from a long-term perspective, and thus explaining migration as an investment. The decision to migrate comes from an analysis of the cost–benefits inherent to migration (Peixoto 2004; Wright and Constantin 2021), so "the decision is conditional upon the return he/she expects to receive from moving in contrast to what is expected from staying" (Korpi and Clark 2017, p. 2).

One macro-level theory is the dual labor market theory (introduced by Piore 1979), which does not accept the neoclassical notion that migration decisions arise from the calculation of individual costs and benefits. This theory suggests that immigration exists because of labor demands in "industrialized or natural resource-rich nations, so the cause of international migration is the pull factors in advanced [...] in terms of needs to do some jobs, which may not be very attractive for the citizens of the host countries." (Wadood et al. 2021, p. 57).

The world systems theory also explains the movement of individuals within a structural view. This theory supports criticism of global capitalism and the spread of neoliberalism, and frames international migration in the supremacy of the Global North (Peixoto 2004; O'Reilly 2023). Migration is seen as one more factor contributing to the perpetuation of the domination of the Third World.

This argument is aligned with theorizing the post-colonial relationship in migration, one approach developed later within contemporary theories providing an understanding of changes in migration and types of migrants, in relation to non-linear, circular, and temporary flows (O'Reilly 2023).

We must also acknowledge the network theory (see Yap 1977). Migrant networks with other migrants living in the host country, as well as family, friends, community, school friends, and office colleagues, make it possible to have a social life and ties in the host country. Within this context, migrants may influence and bring other people from their home country (Peixoto 2004; Wadood et al. 2021).

To advance understanding of human mobility, De Haas (2021) presented the aspirations–capabilities framework. The elaboration is drawn on Berlin's (1969) concepts of positive and negative liberty and on Carling's (2002) concept of involuntary immobility regarding

people who wish, but do not have the ability, to migrate. The framework conceptualizes migration as:

> "a function of aspirations and capabilities to migrate within given sets of perceived geographical opportunity structures. It distinguishes between the instrumental (means-to-an-end) and intrinsic (directly wellbeing-affecting) dimensions of human mobility. This yields a vision in which moving and staying are seen as complementary manifestations of migratory agency and in which human mobility is defined as people's capability to choose where to live, including the option to stay, rather than as the act of moving or migrating itself." (De Haas 2021, p. 1).

### 1.2. Motives Influencing Migration

The decision to emigrate starts in the country of origin as a project constructed before the journey (Basabe et al. 2004). It can be an individual, dyadic, or family project, and if joint, thus involves negotiation (Erlinghagen 2021).

The motives for migrating are wide-ranging and can be explained by the sociological theories outlined in the previous section as linked to personal characteristics and the social and political situation of the country (Fischer-Souan 2019). Among the motives shaping the decision to leave one's country to live in another one are spousal decision, family reunification, more freedom, greater status of the country of destination, marriage, to help people in his/her country, country development, public persecution, crime and corruption, group threat, getting to know new cultures, challenges, education, adventure, supporting family, economic difficulties, seeking employment, and a better life (Basabe et al. 2004).

Financial safety needs have been identified in the literature (e.g., Basabe et al. 2004; Caballero et al. 2019; Dohlman et al. 2019; Fischer-Souan 2019) as a major driver of migration, even for highly specialized professionals with good living conditions, such as physicians. However, for them, as for other highly skilled migrants (Fischer-Souan 2019), factors related to self-actualization, such as the desire for professional development through training opportunities and research, are also major contributors in deciding to go to another country, particularly a higher income country (Dohlman et al. 2019). These motives to migrate from the push–pull model perspective (Peixoto 2004; Wadood et al. 2021) are explained by a comparative analysis between the opportunities for career development in the country of origin with other countries of destination. The potential migrant realizes there are fewer opportunities for career development, thus raising the decision to migrate. However, from the human capital theory perspective, the decision could be explained as a future investment, after balancing the cost–benefits of migrating (Peixoto 2004; Wright and Constantin 2021). At a macro-level—and inspired by the dual labor market theory (Wadood et al. 2021)—a physician and other professionals might migrate because there is a lack of workforce in several countries, such as in Europe and particularly in rural Portugal.

Regarding Europe and Portugal's position in the world, as well as its colonial past, world systems theory and a post-colonial approach may provide an analytical framework to explain migration of third-country nationals to the Alentejo.

Six dimensions of factors explaining migration have been identified in the literature. The most expressive is the economic dimension (e.g., Basabe et al. 2004; Caballero et al. 2019; Dohlman et al. 2019; Fischer-Souan 2019), which can be based on economic difficulties, as well as the desire to have better living conditions for him/herself and his/her family, along with education and health services, aligned with push–pull models. This dimension is followed by seeking stimulation and personal development—that is, new features and opportunities (e.g., Basabe et al. 2004; Fischer-Souan 2019). This perspective seems to be related to the human capital theory. The third dimension is idealism, or the desire to improve and contribute to the development of his/her country and to support its people (e.g., Basabe et al. 2004). The fourth dimension is individual mobility, including personal interests such as the desire to live in a country with more status or being in a transnational marriage (e.g., Basabe et al. 2004; Fischer-Souan 2019). The desire to live in another country, being free and capable of that move is related to De Haas's (2021)

proposal. The fifth is related to family (i.e., family reunification) and thus remains as the migrant network theory. Finally, the sixth dimension is based on an external situation that pushes the migrant to leave his/her country, such as the need to seek political asylum in a safer country (e.g., Basabe et al. 2004), the labor market and social institutions, and norms of discontent, which are perceived as conservative, and intolerant of 'alternative lifestyles' (i.e., discouraging homosexuality; Fischer-Souan 2019). This dimension is related to different perspectives, such as push–pull models, as well as the macro-level and structural perspectives. Complementarily, and focusing on a Portuguese municipality in Alentejo as a case study, Fonseca et al. (2021) presented the spiritual aspects and alternative way of life as an international attraction: "the Tamera Ecovillage, the Community 108 and the spiritual retreat of a community led by guru Mooji are examples of alternative ways of life attracting members from all over the world" (p. 6). According to the aspirations–capabilities framework of De Haas (2021), people may be free, able, and active (and not victims) to decide to migrate and live in other places within given sets of perceived opportunity structures.

In the literature, other terms for categories are also considered, such as (a) preservation: physical, social, and psychological security; (b) self-development: personal growth in abilities, knowledge, and skills; and (c) materialism: financial wellbeing and wealth (Tartakovsky and Schwartz 2001); or (i) demographic, (ii) economic, (iii) environmental, (iv) human development, (v) individual, (vi) politico–institutional, (vii) security, (viii) socio-cultural, and (ix) supranational distinguished by Czaika and Reinprecht (2020) after an assessment of the migration literature. There are also micro- and macro-level factors, including cultural, economic, political, and societal structures as well as individual characteristics (Fischer-Souan 2019).

The motives shaping the decision to migrate to a rural area may be discussed in terms of its attraction regarding the lack of studies in the literature on the topic. Rural areas attract immigrants depending on the availability of employment, particularly in the agricultural sector, where there is a lack of workforce caused by depopulation, and locals are not interested in unskilled jobs and hard work (Sampedro and Camarero 2018). The nature of work has been changing, allowing for remote working, which lets digital nomads move to any place, including rural areas and smaller cities (Gauci 2020). Such places are interesting for international immigrants because of their quiet and peaceful surroundings, which are better for their children (Flynn and Kay 2017).

According to Gauci (2020), smaller cities present advantages for immigrants seeking to move to those regions, because it is easier to create a social network with the host society; there is, for example, less risk of structural segregation in schools, in part because there may only be one school in the town. There is also a greater chance to interact with host communities, providing a closer and tight-knit safety network and a greater role for local organizations and community leaders. Local employment, better housing prices and satisfactory connections to larger cities with employment opportunities are additional benefits of rural areas.

Finally, we highlight that "there is growing acknowledgement that migration is not the outcome of a single factor or a 'root cause' but of complex configurations of multiple, interdependent and interacting factors" (Migration Data Portal 2021). Migration drivers influence the decisions to migrate (International Organization for Migration (IOM) (2019)), as well as shaping the flows, routes, and the attractiveness of a location (Czaika and Reinprecht 2020).

### 1.3. What Is Attractive about Portugal?

Historically, Portugal has been a territory of population flux. Migration flows have had different speeds and intensities since the 15th century, and this variance has been associated with demographic, economic, and social changes (Góis 2023; Góis and Marques 2018; Padilla and Ortiz 2012). Although immigration to Portugal has been fairly consistent, there has recently been a boom in such migration (Oliveira 2022; Padilla and Ortiz 2012; Pordata

2021). This increase could be based on the fact that the European Union is receiving more and more immigrants from third countries; in 2022, there were 23.8 million migrants, 5.3% of the EU's 446.7 million inhabitants (European Commission 2023a). However, the Portuguese case seems to be unique, according to Góis (2023), because of a set of particularities:

1.  Its geographic position at the convergence of different migration systems has led to flows to Europe and in Europe, as well as from and to other continents.
2.  Its colonial past has prompted migrants from Portuguese-speaking African countries (e.g., Angola, Cape Verde, Guinea Bissau, and Mozambique) and Brazil.
3.  The evolution of its demographic structure, a significant ageing population, and a constant negative natural balance have made the country dependent on migration to sustain its population.
4.  Its accession to the European Union, and therefore to its migration policy—namely access to the Schengen agreement, which influences inflows—has made it part of a region that is one of the most attractive places in the world for migrants (ESPON 2018).
5.  Finally, its sociopolitical evolution in the last few decades have attracted migrants, in part due to its open-door migration policy.

Successive amendments to the Foreigners Law (Law no. 23/2007, of 4 July, in its current 13th version) and the Nationality Law (Law no. 37/81, of 3 October, in its current 11th version) have encouraged migratory flows to Portugal. Compared to other European countries, Portugal is more competitive. It allows a migratory path into Europe, which would make it possible, in a few years, to move to another European country. In the words of Góis (2023), Portugal provides a European passport and increases the number of immigrants as well as nationals. Facts and figures related to immigrants in Portugal and their distribution are thus relevant to a better understanding of the reality of migration in Europe and how they can sustain smaller regions (Natale et al. 2019).

### 1.4. International Migrants and Rural Alentejo

Rural areas have increasingly begun to attract foreigners (Collantes et al. 2013), and the Alentejo is an example of this shift (Figure 1). This region is the subject of a current project examining its socioeconomic and demographic challenges, as it is one of the most depopulated regions, marked by ageing and low economic development. Scholarship must therefore support local policymakers by providing studies (European Commission 2023b). It is also a destination that has not, historically, had international migrant populations. Rural Portugal has experienced a delay in its attraction for agri-business investors, compared to other Southern countries in Europe, except for Odemira, a rural municipality made up "of a cobweb of connections that reaches out to several continents, permanently being remade by actively taking part in global commodity chains, labour and amenity migration, and commodification of natural resources" (Fonseca et al. 2021, p. 2).

In 2021, there were 23,737 (19.8%) newcomers (Serviço de Estrangeiros e Fronteiras/ Gabinete de Estudos, Planeamento e Formação (SEF/GEPF) 2023). In total, there were 52,316 immigrants of 1,089,023 in the country (Oliveira 2022; Pordata 2021). The most represented nationalities among non-European Union citizens in the Alentejo were Brazilian (10,083), Indian (7383), Nepalese (3659), Ukrainian (2224), Chinese (1591), British (1434). and Angolan (723; Pordata 2022).

In the Alentejo region, the population density is around 22 inhabitants/km$^2$ (Pordata 2023), which is much less dense than the country as a whole, at 113 individuals/km$^2$. Lisbon, the capital, by way of comparison, presents 952 inhabitants/km$^2$ (Instituto Nacional de Estatística (INE) 2023a). There was a decrease in inhabitants in 2021, around 7.0% or 704,533 people (Pordata 2023). However, the Portuguese municipality, Odemira, has been at the top of the population growth in the last decade, based on international newcomers working in agriculture.

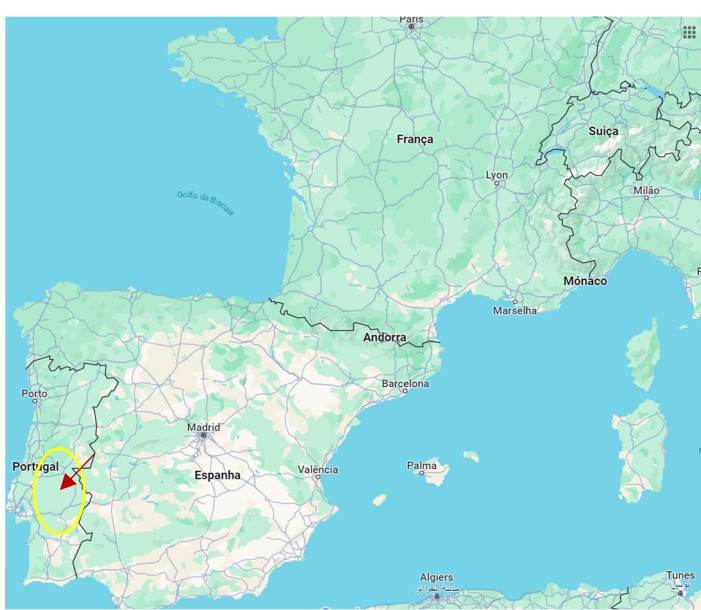

**Figure 1.** Map of the Alentejo region in Portugal, Europe. Source: Google Maps. Available online: https://www.google.pt/maps/place/Alentejo/@42.8177341,-4.1164962,6z/data=!4m6!3m5!1s0 xd1b7119e7099da1:0x967a8ef2a673a03f!8m2!3d38.2003808!4d-7.800085!16s/g/128dgx6tm?hl=pt-PT&entry=ttu (accessed on 10 March 2024).

The ageing index is not surprising in the demographic overview. It is high and becomes higher near the border with Spain (Mauritti et al. 2019). The population is ageing, which means there are 219 elderly people per 100 young people, and the renewal of the population is decreasing (Pordata 2023). The birth rate is 7.4, compared to 9.3 in the Metropolitan Area of Lisbon and 7.7 in the whole country (Instituto Nacional de Estatística (INE) 2023b).

The Alentejo also has the most derelict houses. Rents, at EUR 328/month, are below the national average and are much lower than those in Lisbon, where rents can be above EUR 1000/month (Instituto Nacional de Estatística (INE) 2023a). Looking at other economic indicators, the results are also poor. The activity rate of the region was the lowest in the country (44.6%); this is based on commerce, industry, agriculture, construction, and financial and state sectors. The region also has the country's highest unemployment, with 4,193,900 (40.5%) unemployed (Instituto Nacional de Estatística (INE) 2023a).

For both national citizens and international immigrants, the Alentejo has a lower crime rate and is a safe place to live. However, human trafficking is present, along with labor exploitation, both of which tend to occur in the agriculture sector (Sistema de Segurança Interna 2022).

This research is supported by demographic and socioeconomic issues of rural regions, namely by the findings of Góis (2023), who highlights the impossibility of natural population renewal. He therefore advocates for international immigration inflows to sustain these regions. Fonseca et al. (2021) also state that international migration is one of the main driving forces of change in rural areas. What is consistent in the literature (Butkus et al. 2018; Diogo 2024; Diogo et al. 2023; Gauci 2020; Quintino 2018) is the focus on rural localities in different countries. McAreavey and Argent (2018) noted that migrants are "evidently important for rural and regional communities. In a Scandinavian context immigration has become a major source of population increase. In other places, labour migrants have boosted and rejuvenated the local population, helping to sustain services and regenerate the economy" (p. 148).

Witnessing the Alentejo, Castro (2011) categorized immigrants as human capital for the region: producers (as businesses and the offer of new goods and services for locals); consumers (e.g., of goods, services, and properties); rehabilitators (of heritage as inhabited houses); as well as their demographic and sociocultural impact (as volunteers, exchanging

experiences). Diogo et al. (2023) found that the increase in immigrants determined the return of public transport and primary schools that were discontinued, as well as the rising investment in real estate and new services (restaurants, hair salons, convenience stores, and so on). Carvalho (2021) underlined that although the region became more competitive after the building of river Alqueva dam, international migrants are essential for the growth and competitiveness of the agricultural sector in Portugal. The sector is unable to attract natives, namely due to the income being significantly lower than the average wage in the economy. Thus, Carvalho (2021) called for rapid policy action.

Knowing why immigrants want to live there is mandatory in supporting policymakers' decisions to attract more people to the territory. This study explores the motives shaping immigrants' intentions to move to the Alentejo, a depopulated region in Portugal, using qualitative methods. Scholars have studied motives to migrate in general, but little attention has been paid to specific motives to go to rural areas and smaller cities that are disinvested, with more challenging and adverse conditions. No studies about the Portuguese reality were found in the literature review conducted. This study therefore sought to address the following research question: What motivates immigrants in deciding to move to rural regions such as the Alentejo in Portugal? Regarding the aim of this study, hypotheses and research questions emerged from the literature on theories explaining migration and factors influencing the movement to other destinations.

This study applied the recommendations from the literature on innovation in research on the migration topic by involving immigrants' participation, which is a complementary way of empowering and including them (Goodson and Grzymala-Kazlowska 2017). This study contributes to the compliance commitments to the achievement of the Sustainable Development Goals (SDGs) of the agenda 2023, and consists of a relevant project, funded by the Asylum Migration and Integration Fund (AMIF) and awarded with the Research, Innovation and Territories Award 2023 by the Iberian Center of Studies. The wide-ranging project has a set of components that includes training for practitioners, sharing practices and scientific research.

## 2. Materials and Methods

The present exploratory study is framed in a significantly extensive co-funded project, so the empirical research design (Blaikie 2010), with a qualitative approach methodology, is transversal to the project. The primary aim is to explore the motives shaping international migrants' intentions to move to the Alentejo, a depopulated region in Portugal. This research aims to contribute specifically to explain migration to a rural and disadvantaged place and to identify motives for a non-EU citizen to decide to live in Portugal in general and in rural Portugal in particular. Supported by the literature, we had a couple of assumptions, which we tested empirically in the current study, examining the data in relation to three hypotheses and searching for patterns and themes, as well as exceptions and contradictions, to support or refute them. Our hypotheses were:

**Hypothesis 1.** *International migration to rural areas, such as the Alentejo region, can be explained by the contribution of the global migration theories of sociology, such as the push–pull model; human capital theory; the dual labour market theory; world systems theory, network theory, and the aspirations–capabilities framework.*

**Hypothesis 2.** *Motives for a non-EU citizen to go to rural areas extend beyond agricultural employment and economic reasons, as consistently addressed by the literature focused on migration in rural areas (e.g., Flynn and Kay 2017; Gauci 2020; McAreavey and Argent 2018).*

**Hypothesis 3.** *Non-EU migrants living in the rural Alentejo region decided to move there influenced by the idea of greater opportunities in Europe.*

*Procedure*

Before the semi-structured interviews, a protocol with a set of questions was prepared based on the literature review and technical reports. The sampling was then tailored according to theoretical guidance (Flick 2005, 2013) and using two criteria: maximum variation (e.g., age, gender, origin, culture; Flick 2013; Goodson and Grzymala-Kazlowska 2017) and convenience. Individual and joint interviews (*n* = 20) took place in 2020 and 2021. They lasted around 1:15 h each and were recorded by informed consent. Although interviews were in person according to participants' availability, due to the COVID-19 pandemic, remote interviews (*n* = 6), online via the Colibri-Zoom platform, were necessary in certain cases. Questions were based on the process of leaving the origin country; migrating, arriving, and integrating in the host country; and practices, along with suggestions to improve local policy and practice.

Participants (*n* = 23) were practitioners (*n* = 8) and international migrants (*n* = 15) in the Alentejo region. Practitioners were working in NGOs for immigrants, those offering a response called Centros Locais de Apoio à Integração de Migrantes (CLAIM; a total of three existed in the territory selected by the project) and in municipalities, those having relevant experience on international migrants' management (considering the number of immigrants and social projects to support them). The expertise of practitioners is relevant, and the discussion on the topic made them re-examine their role in policy and practice efforts in response to the crisis (Rine 2018).

The practitioners, three males (37.5%) and five females (62.5%), had a higher education diploma (except for one). They were all born in Portugal.

The second group of participants were newcomers from third countries living in the Alentejo region. The access to immigrants was provided by the referred entities, as well as by the immigrants interviewed (i.e., snowball sampling technique), selected by maximum variation supported by previously given information (e.g., age, country of origin, education, length of stay in the host society, household, and professional situation). Data were confirmed in the first contact with immigrants, and details were explored. The purpose of this study was also explained; migrants were asked if they were interested and, if so, an interview was scheduled. Immigrants' participation in research on migration is recommended by the literature on innovation in research on the topic to avoid an ethnocentric vision and to empower the immigrants themselves through participation and co-production of knowledge (Goodson and Grzymala-Kazlowska 2017) within a participatory approach (Heckmann 2008).

The interview participants, eight males (53.33%) and seven females (46.67%), between 26 and 67 years old, were native to Brazil, Cape Verde, China, Guinea-Bissau, Moldova, São Tomé and Príncipe, Ukraine, and Venezuela. Their education level ranged from secondary education to a bachelor's, master's, or PhD degree. They worked in academia, accounting, factories, hairdressing, social services, and security. The length of stay in Portugal ranged from 4 to 30 years—thus, from 1993 until more recently in 2019. When asked about the future, six (40%) answered with their intention to return to their home country and eight (53.33%) said they would remain in Portugal; one (6.67%) declined to answer.

Interviews were transcribed verbatim in the conversation language, Portuguese (except for one of them), using Express Scribe Pro software version 10.05.

Thematic analysis (Braun and Clarke 2006) was conducted, supported by MaxQDA2022 software to ensure a coding process that was organized, coherent, and consistent. The transcription files were uploaded to the software. After reading the transcripts, listening to audio recorded files a second time, and reviewing the field notes taken by the authors, the next step was conducted. Empirical statements transcribed were then coded and categorized (e.g., cheap rent; suitable education for their children; promise of work; support from a local organization). The interpretation of the narratives captured participants' experiences and meanings; trends and patterns were identified. The results were then analyzed to draw conclusions to answer the research question. During the entire process, maps, diagrams, and insights (initial and advanced) were written to support the analysis. Finally,

to write this article, quotations were translated into English by the author and revised by an English speaker.

## 3. Results

The findings present the motivations shaping immigrants' intentions to move to the Alentejo region, a depopulated region in Portugal, which is a low-density territory. This section centers around two topics, the attraction (a) to the country (Portugal) and (b) to rural areas and smaller cities (i.e., the Alentejo). At the end, the results are presented in Figure 2.

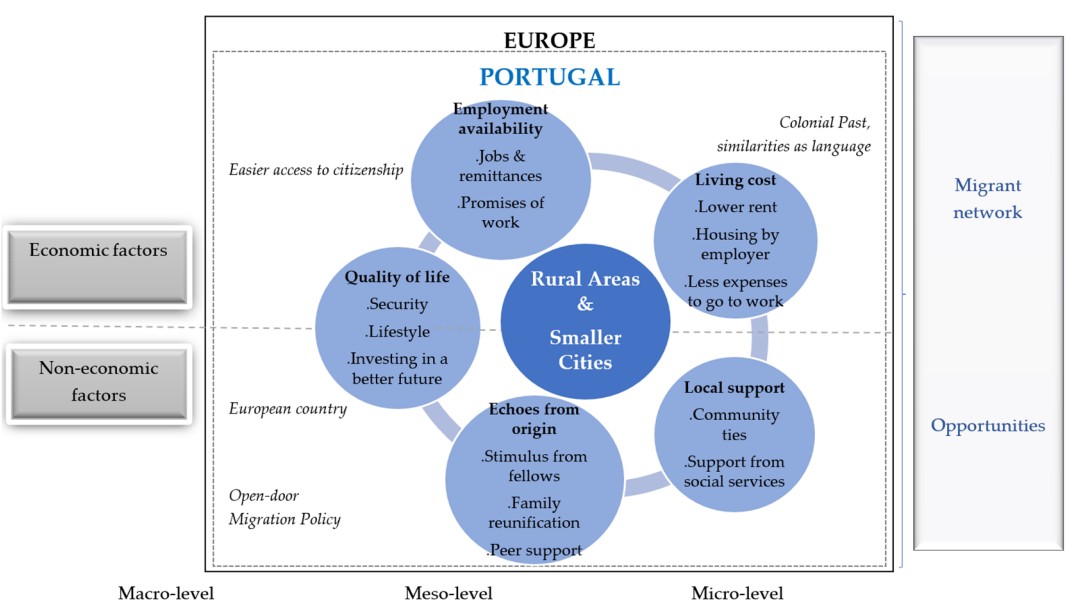

**Figure 2.** Factors for moving to Portuguese rural areas and smaller cities. Source: author's own elaboration.

### 3.1. Attraction of Portugal to International Immigrants

The motives for international migration, particularly for citizens of non-European countries, to Portugal can, in general, can be subdivided into four dimensions: a Portuguese open-door migration policy; easier access to citizenship; it being a European country; or the colonial past. Portugal is well known for its open-door migration policy, considering its lack of workforce and low birth rate. Compared to other European countries, it is easier to become a legal citizen in Portugal (i.e., obtain authorization to stay permanently).

> "procedures to become regularized are quicker. It may take three or four years, but he/she knows that he/she will get an authorization. In 12 months, he/she will get a residence [. . .] it is less complex than in Italy, Spain or Germany." (Social worker 2, male, Municipality B).

Once in Portugal, they may obtain Portuguese citizenship in a few years—and therefore European citizenship—which is very attractive for non-European immigrants.

> "I always say to them, the first thing to do is think about having a baby. Because having a baby means request by their children." (Social worker, female, CLAIM 1).

Portugal is a European country, and its health services are superior, often significantly better than in the country of origin. A similar mindset is also apparent for education.

> "A migrant told me that she wanted her little girl to have the opportunity of a European education and to access health services. She means that here she could give them what she could never provide in her home country." (Social worker, female, CLAIM 2).

Finally, participants referred to the large number and increase in Brazilians in Portugal, both in the past and at present, because Brazil was a Portuguese colony and therefore the same language is spoken, which facilitates integration. Inflows of Brazilians are permanent; however, several milestones are pointed out.

"It is always the Brazilians!"

(Social worker, female, CLAIM 3).

"And what is there for the Brazilians? [many of them in Portugal are over-qualified]". (Immigrant, female, Brazil).

*3.2. Attraction of Rural Areas and Smaller Cities*

Economic and non-economic factors support the decision of international migrants to go to rural areas and smaller cities. The economic reasons include employment for remittances, as well as lower rents, living costs, and daily expenses than in medium and bigger cities. Non-economic reasons include security, quality of life, informal network support, and a better future perspective for themselves and for their children. Five dimensions are discussed below.

3.2.1. Employment Availability

Economic reasons are the most relevant aspect for immigrants related to employment, along with remittances to their family still in the country of origin. There is a labour shortage in these regions with declining populations, which has created job vacancies. Immigrants are attracted either by job advertisements, word of mouth, or promises of work with high salaries. The agricultural sector hiring international immigrants is commonplace; however, there are other sectors recruiting many immigrant workers, such as factories, the construction industry, cleaning industry, tourism, and social care services. Low-paid and arduous farm work may be not attractive to some migrants, who tend to seek employment in other sectors. These economic sectors demand lower educational levels and lower language skills.

"It was the employment, and a respectable salary. I think that a person who leaves his/her home for another one, his/her dream is to grow and develop". (Immigrant, female, São Tomé and Príncipe).

"Here, if the two members of the couple are employed, they have guaranteed salaries at the end of the month and are able to pay bills and buy food". (Immigrant, female, Moldova).

"They are focused on employment, for sending remittances to family to achieve better life conditions". (Social worker, female, CLAIM 3).

"Beja nowadays has a lot of good things, for example a lot of agriculture; that's the reason why there are so many immigrants, because of employment." (Immigrant, female, Ukraine).

"[...] residential care for the elderly, cleaning, companies that opened doors to international immigrants, such as Tyco, Quemiche and Embraer [...] here, it is not so demanding". (Immigrant, female, Brazil).

"[the capital] Lisbon is an urban center where the Portuguese language is needed. Thus, the only way to find a job is leave, and they [international migrants] come to these regions a lot". (Social worker, female, CLAIM 3).

The pursuit of workers with higher education was also mentioned:

"Immigrants who come for a specific company with a previous promise of work, is more common for active employees in a specialized field of work." (Immigrant, female, Brazil).

These regions offer seasonal jobs and employment in agriculture that can be attractive, or not, for some of the international labour force,

"In the Alentejo, work in agriculture is temporary". (Immigrant, male, Guinea Bissau).

"They [international immigrants] work temporarily on the olive harvest, then go away, then come back, then go to pick red fruits". (Social worker, female, municipality).

### 3.2.2. Living Costs

Rental properties in rural regions have lower prices than in bigger cities, so the proportion of the cost of housing in a household's budget is lower. That is particularly relevant for lower income families. Receiving the minimum wage, it is still possible to satisfy basic needs, pay for daily expenses such as housing and food, and to maintain quality of life.

"The city here is really calm and the cost of living is cheap". (Immigrant, male, Guinea Bissau)

"The rent is lower than in Lisbon, for example". (Immigrant, male, Cape Verde).

In a region marked by a shortage of housing for those who work in agriculture, employers may offer housing on their own property or nearby for newcomers. The benefit is likely to be present before coming to the region and included in the promise of work. That may represent a reduced housing cost and no travel costs to work.

"[...] there are excellent examples of companies offering both an excellent accommodation service and employment [...]" (Social worker, male, municipality).

"The majority work on farms, thus a lot of them live in the own farming village". (Psychologist, male, municipality).

In any situation, once living and working in a smaller city, expenses to go to work (among others) are less than in bigger cities:

"It is the cost of living, the quality of life offered here [...] they [immigrants in a big city] earn to survive, and not to live! To survive..." (Social worker, female, CLAIM 3).

### 3.2.3. Quality of Life

Security and the sense of peace was identified as an attraction for everyone in this region. This was consistently stated by Brazilians, because there are many crimes and homicides in their country of origin, particularly in bigger cities.

"I walk on the streets fearless [...] safely, any time day and night." (Immigrant, male, Brazil).

"I live with quality. I live as any other citizen. If I want to go out for dinner with my grandchildren, I go. If we want to go to the beach, we go [...] visit another city and a museum, we go. These are things that were not accessible to me in my country of origin." (Immigrant, female, Brazil).

"Here... there is a tranquility... unhurried, in contrast to the city... quality of life..." (Immigrant, female, China).

Investing in a better future is central for the migrants. A relevant dimension is education and health care services, as well as career development. Quality education and health care tend to be inaccessible in the migrants' country of origin.

"Here you may have your family, you can provide your children with an education... there is a hospital, a proper hospital, as well as the local health center. It is same for education: there are proper schools, and there is even proper higher education." (Immigrant, male, Brazil).

"She [an immigrant client] told me, 'I intend to provide a European education to my children. I would like my children to have health care access. I would like

them to be given opportunities impossible to get in my home country'". (Social worker, female, CLAIM 3).

### 3.2.4. Local Support

Local NGOs specializing in migration are seen as safe and reliable services, providing social and financial support for immigrants if needed. However, these organizations are sought-after by newcomers to apply for residence authorization in Portugal. Residence authorization is fundamental for regularization and thus for the right to sign an employment contract and have labour stability, as well as for social integration. In fact, even before arriving in these regions, immigrants acknowledge and request services from these local NGOs, guided by migrant networks and by word-of-mouth dynamics. According to one participant, a smaller region may make the difference by reducing the waiting time. Counting on this formal support, immigrants feel confident to go to an unknown and remote place. Thus, quality services and strong professional relationships with social workers represent an attraction.

> "This is the easiest place to get documents. Thus, immigrants come here from other cities to request Caritas support, to apply for documents." (Immigrant, male, Guinea Bissau).

> "I used to attend six clients in a week, and nowadays I attend six, but in one day. [. . .] Perhaps, 90% or nearly 100% are clients from outside the city." (Social worker, male, CLAIM 1).

> "Whenever close relationships are the practice, they [immigrant clients] get a reference here. We become someone that they know, they can count on, and then they request our support always, even for other colleagues and other services." (Social worker, male, municipality).

Similarly, community ties previously established—or if referred by someone else by word of mouth—lead to more empathy and the desire to go to that place. Close relationships with locals allow intercultural communities and improve wellbeing once diversity is present.

> "So, you start becoming friends [with locals]. . ." (Immigrant, female, Brazil).

> "Portalegre is a calm place that has generous people; they receive us with a warm welcome that not all cities would do." (Immigrant, male, Guinea Bissau).

### 3.2.5. Echoes from Country of Origin

Fellow immigrants from the country of origin stimulate motivation to go to the place where they live, particularly once positive features and benefits about the externalities of the territory are highlighted. The informal network that is then established is also relevant in providing peer support, considering the absence of extended family. Provision of social and cultural support as well as child supervision are relevant factors in deciding to go to a new place.

> "Immigrants appeal to a lot of friends. And if here are jobs, they will come and stay". (Immigrant, female, Brazil).

> "[Immigrant] people come to here, and then one after another". (Immigrant, female, Brazil).

Family reunification is an opportunity for immigrants in rural areas and smaller cities, because there are jobs vacancies, lower cost of living, security, quality education, formal and informal support, and other conditions.

> "They [international migrants] aim to stay here, bring their own family, as wife and children that stay in their home country [. . .] men want to come here, get a job and then bring their family." (Social worker, female, CLAIM 2).

Participants also presented suggestions to improve the attraction of rural areas to international migrants that were very focused on infrastructural needs, such as greater accessibility

to the local airport and creating high-speed motorways; as well as improving the offerings for public transport; and increasing the number of available houses through the renovation of old ones, because there is a lack of housing availability, although a large number of dwellings are empty or not habitable.

> "Public transport systems should be guaranteed, housing and. . . social justice, because there is not social justice here, yet!" (Immigrant, male, Venezuela).

> "We have here an issue about the airport that does not operate, the train does not operate. Therefore, while we don't reinforce local infrastructure, the region and around will be hampered." (Psychologist, female, municipality).

> "[. . .] the houses that belong to the municipality are already occupied. If they were renovated, if there was this kind of investment. . . I mean there are manifold derelict and inhabited properties in the old town, a lot of them. . ." (Social worker, female, CLAIM 3).

In addition to the dimensions explored above, the results also suggest that migrant networks are a push determinant and support, and there is also motivation to pursue new and greater economic, social, and educational opportunities outside the country of origin, which were constantly mentioned in participants' narratives.

Finally, the results present multilevel factors—micro-, mezzo-, and macro-level influences—for international migrants to want to go to rural areas and smaller cities. The micro-level factors included migrating as an individual choice, based on economic and non-economic reasons for him/herself and his/her family. Mezzo-level factors included family and peer ties, the migrant network, and community relationships. The macro-level factors involved policies, the major labour market, and other dynamics.

## 4. Discussion and Conclusions

Studying international migration remains relevant; according to O'Reilly (2023), it has the potential to transform individuals and societies in varied and remarkable ways, enriching, and stimulating competition and transformations, as desired for the sustainability of rural regions.

The present article explored factors shaping international migrants' intentions to move specifically to rural areas and smaller cities such as the region of Alentejo, in Portugal. It is an innovative study, because there has been insufficient research on migration to rural places, with even less research on this region and none specifically about the motives to move there. The results are intended to inform policymakers as they design evidence-based plans to attract international migrants aware of their value to affect sustainable local development (Butkus et al. 2018; Diogo 2024; Diogo et al. 2023; Gauci 2020; Quintino 2018).

The findings present a set of consistent factors for a person deciding to move to this rural region, with these being underfunded and more adverse places, with multiple challenges (Carvalho and Oliveira 2017; ESPON 2018; Morén-Alegret et al. 2021; Mota 2019).

On the one hand, the results are aligned with the literature. The most expressive motive to move to the rural Alentejo was economic (as for example noted by Basabe et al. 2004; Caballero et al. 2019; Dohlman et al. 2019; Fischer-Souan 2019), as well as the desire to have better living conditions, education, and health services. Seeking stimulation and personal development (e.g., Basabe et al. 2004; Fischer-Souan 2019) were present regarding the desire for better opportunities for their children. Europe, and inherently the Portuguese Alentejo, is perceived by participants as a place of opportunities; thus, to come for those opportunities and to bring the family. In addition, it is seen as a place with more status (e.g., Basabe et al. 2004; Fischer-Souan 2019).

On the other hand, seeking political asylum; the desire to improve the development of the country of origin (Basabe et al. 2004); spiritual aspects; and pursuing an alternative way of life (Fischer-Souan 2019; Fonseca et al. 2021) were not presented by the participants as relevant in their move to the region.

This study presents specific motives for the desire to live in rural regions such as the Alentejo that are not observed in other destinations, such as big cities. A contribution of this study is that migrants decided to move to these regions for multiple factors that are associated with a higher quality of life and lower living costs in these regions than in bigger cities. These features may be used by politicians through initiatives to apply for the interest of international newcomers, namely those from insecure and dangerous places.

Other findings distinct to this study are that social workers of local NGOs for the integration of migrants may be a determinant to attract newcomers. Social workers provide information, support, and a source of reliance, even for those migrants who are living outside the region, since they are recommended as a major asset by other migrants. Gauci (2020) noted less difficulties in smaller places to create a social network with the host society but did not find any evidence about local professionals' roles.

Based on the assumption that international migration to rural areas, such as the Alentejo region, can be explained by the contribution of the global migration theories of sociology, such as, for example, the push–pull model; human capital theory; the dual labour market theory; world systems theory, network theory; and the aspirations–capabilities framework, Hypothesis 1 is supported by the empirical data. Thus, the set of dimensions among the motives found for international migrants to move to these places, such as the Alentejo, are not substantially different from motives for general migration as suggested by the theories briefly covered above.

The results are clearly based on multilevel factors: micro, mezzo, and macro. Although classic sociological explanations of migration simply propose two wide groups: based on micro-sociological causes, such as an individual choice, and on macro-sociological causes, such as the result of social strengths. The third level group, framed as the mezzo causes, is less mentioned in the literature. However, the findings of this study dovetail it with the importance of the social services relationship, community ties, the peer group, and family reunification. These results align, in a certain way, with the discussion of Wadood et al. (2021) about approaching mezzo factors such as family ties, social networks, peer groups, and community bonds.

This study thus suggests that rural areas and smaller cities are not significantly different from medium and big cities when considering sociological approaches to the phenomena of migration.

Firstly, the results confirm the push–pull model and human capital theory (Peixoto 2004; Wadood et al. 2021) as explanations based on individual motives to go to live in the Alentejo region being rooted in economic will. The economic situation in the country of origin is forcing people to leave their country; however, the Portuguese Alentejo demands a labour workforce and offers better conditions and livelihoods. This includes the long-term perspective of migration as an investment as explained by the human capital theory (Korpi and Clark 2017; Wright and Constantin 2021); thus, migrants come to the Alentejo seeking future opportunities for their children, namely though better health care services and schools.

Regarding macro-level theories, such as the dual labour market theory and world systems theory, migration decisions do not arise from individuals, but because of global capitalism, labour demands (Wadood et al. 2021), and the domination of the Third World (O'Reilly 2023). The post-colonial relationship in migration is present in the results of this study. The results present the labour shortage in the Alentejo, particularly in the agricultural sector that presents a recent investment based on the water offer. However, the depopulation influences other sectors recruiting a large number of workers for the economic development of the region and the country. Portuguese former colonies such as Brasil are the most represented countries of origin because of its past relationship with Portugal. Brazilians in a counter movement are supporting the economic development of the contemporary Portugal.

Brazilian inflows among others corroborate the migrant networks as well (Peixoto 2004; Wadood et al. 2021) as a migration explanation in the Alentejo. They influence and bring

other people from their home country, presenting the externalities of the Alentejo territory. Networking with fellow immigrants was consistently and unanimously mentioned by the participants that may in fact denote Alentejo as an unknown destination in Brazil, since it is a new immigration destination (McAreavey 2017).

Later contemporary theories are also confirmed regarding changes in migration and types of migrants (O'Reilly 2023) as the aspirations–capabilities framework presented by De Haas (2021) highlighting that people may be free and able to decide to move and where to live, namely if they prefer a European culture and education for their children.

The second hypothesis, based on the suggestion that the motives for a non-EU citizen to go to rural areas extend beyond agricultural employment and economic reasons, is confirmed by the empirical results as well. Thus, several dimensions of factors influencing immigrants emerge from the findings, particularly employment availability for remittances and promises of work before going to the region, and lower living costs, namely rents and expenses to go to work, compared to bigger cities. However, this study found other factors were also important, such as quality of life, particularly security and prospects of a better future, human capital, and access to quality education and health care services; local support based on close professional relationships with social workers and community bonds; and echoes of the country of origin in terms of family and peers. Some similarities with previous studies about general migration are also present (Basabe et al. 2004; Caballero et al. 2019; Dohlman et al. 2019; Fischer-Souan 2019). These include the approachability of social workers and community bonds, living costs, and security from crime and robberies, all of which are more evidence for going to rural areas than for general migration, as the Alentejo has less demand for support, as well as lower prices and rates of crime (Sistema de Segurança Interna 2022). Marriage with locals was not mentioned as a factor of attraction (Basabe et al. 2004; Gauci 2020).

The previous literature has focused at length on better income-earning prospects and economic reasons (e.g., Basabe et al. 2004; Caballero et al. 2019; Dohlman et al. 2019; Fischer-Souan 2019) to explain the increase in immigrants in the Global North. However, this study balances the economic and non-economic aspects of the attractiveness of rural areas and smaller cities. For Brazilians, for example, security is fundamental and associated with survival. The lifestyle and prospects of a better future for the migrant and his/her children represent a significant weight for determining to leave their country and move to these regions, in accordance with human capital theory. Peer stimulus and support, relationships with practitioners, and community bonds also seem to be as important and encouraging as getting a job or promise of work, along with a lower cost of living, as Gauci (2020) highlights in a discussion of property (rental) prices.

To finish, Hypothesis 3 assumed that non-EU migrants living in the rural Alentejo region decided to move there influenced by the idea of greater opportunities in Europe, and this is supported by the empirical results of this study. This means that a complementary and relevant aspect for a person wanting to move to a rural region in Portugal is that the Alentejo region is in Portugal and thus in Europe.

Europe is perceived as powerful and wealthy nation-states that explain international migration according to the world systems theory (O'Reilly 2023). Another perspective is based on the aspirations–capabilities framework of De Haas (2021) who achieved a "more meaningful understanding of agency in migration processes by conceptualizing migration as a function of aspirations and capabilities to migrate within given sets of perceived opportunity structures" (pp. 30–31). Regarding the view of the author, human mobility is explained by people's freedom or ability to decide the place to live, for example, as in Europe.

We acknowledge that Portugal is often not the first option as a destination for many newcomers (Castro 2011). The attraction of Portugal, as stated in the literature (Góis 2023), is present in participants' narratives, such as having the opportunity to obtain Portuguese citizenship and thus European citizenship, which ultimately enables free movement in Europe. The open-door Portuguese migration policy (Law no. 23/2007, of 4 July, in its

current 13th version) and citizenship access policy (Law no. 37/81, of 3 October, in its current 11th version) seem as though they will continue to provide increasing inflows of non-EU citizens to Portugal and then to other European countries. The Portuguese integration policy is well-positioned by the Migrant Integration Policy Index (Solano and Huddleston 2020); however, special attention should be paid to its management—that is, to local migration policies.

The attractiveness of Portugal, as presented by Góis (2023), explains the facts and figures about the distribution of the most-represented countries of origin in rural Portuguese areas, as previously detailed. The massive migration of Brazilians and Angolans is supported by the colonial past of Portugal and bilateral relationships of cooperation. Again, as in world systems theory, theories on the post-colonial relationship in migration (O'Reilly 2023) implies instituted relations of inequality.

As referenced, the huge numbers of Brazilians and Asian citizens may also be explained by network theory. Migrants may thus influence and bring other people from their home country to the rural Alentejo region (Wadood et al. 2021).

The European Action Plan on Integration and Inclusion (European Commission 2020) presents strategic actions covering all the different stages and phases of the integration process: pre-departure measures, reception and early integration, long-term integration, and the building of inclusive and cohesive societies. We claim that national, regional, and local governments must be aware of this plan, and informed by evidence, provide their own local plans. These should be created within a participatory and multiway process, involving all stakeholders: migrant representatives, social services, employers, policymakers, and communities. The evidence-based management of migration is essential for sustaining development and creating an inclusive society (European Union 2023; United Nations 2015). Migration management, at multiple levels—i.e., at the macro-level shared between member states and at a micro-level by local policy (Gauci 2020)—may strategically disperse international migrants for 'in need' regions within a sustainable development vision within the frame of the SDGs, because immigrants play an important role in sustaining communities (Collantes et al. 2013; Sampedro and Camarero 2018).

There are also other recommendations for policy and practice. Policy should consider (a) stimulating housing availability from employers and from the rental market by fiscal support measures to renovate old houses; (b) supporting employers recruiting and integrating an international workforce, as well as monitoring and supervising it to create confidence on both sides, a win–win process that would allow the hiring of more workers and stable employment; (c) reactivating public transport offers—airports and trains; and (d) involving stakeholders in local plans and actions, namely the host community and immigrants, for inclusive societies. In practice, it is necessary to be approachable and available for everyone via any communication channel, for newcomers and those currently living elsewhere. Finally, for both policy and practice, evidence-based decision-making is essential.

This study has certain limitations, which include the small sample size; however, the research as designed, with a qualitative approach, did not intend representativeness, and theoretical saturation was assured. Further research should focus on the role of amenities that determine the attractiveness of rural regions in Europe for migrants, as well as the specific impact of international migrants in these regions to inform policy.

**Funding:** This research is part of the project called Ir Além–A Inclusão Social de NPT e o Desenvolvimento de Territórios de Baixa Densidade, co-funded by the Asylum Migration and Integration Fund, operation number PT/2020/FAMI/535. The project was awarded by the Centro de Estudos Ibéricos with the Research, Innovation and Territories Award 2023. This work also was support-ed by national funds through the Fundação para a Ciência e a Tecnologia, I.P. (Portuguese Foundation for Science and Technology) by the project UIDB/05064/2020 (VALORIZA - Research Centre for Endogenous Resource Valorization).

**Institutional Review Board Statement:** Ethical review and approval were waived for this study due to it is not mandatory to have ethical permission from the Ethical Commission. The Ethical

Code of Instituto Politécnico de Portalegre: https://www.ipportalegre.pt/pt/sobre-nos/qualidade/sistema-de-gestao-da-responsabilidade-social/etica/ (accessed on 29 November 2023) and https://files.diariodarepublica.pt/2s/2021/08/157000000/0026900277.pdf (accessed on 29 November 2023).

**Informed Consent Statement:** Informed consent was obtained from all subjects involved in the study.

**Data Availability Statement:** Data are available on request.

**Acknowledgments:** We acknowledge the administrative and technical support from the research and innovation office, and from the whole Polytechnique University of Portalegre; and finally, the collaboration of volunteers, research fellows, and partners. Our main words are to the immigrants and practitioners who were available to participate in the present research.

**Conflicts of Interest:** The author declares no conflicts of interest.

## Note

[1] A person who moves away from his or her place of usual residence, whether within a country or across an international border, temporarily or permanently, and for a variety of reasons (International Organization for Migration (IOM) 2019). This study focuses primarily on non-EU migrants, as it is co-funded by the European Union Asylum Migration and Integration Fund (AMIF).

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
