# Peer review of "“Why Here?”—Pull Factors for the Attraction of Non-EU Immigrants to Rural Areas and Smaller Cities"

_socsci, doi:10.3390/socsci13040184_

Round 1

Reviewer 1 Report

Comments and Suggestions for Authors

This is an article that analyzes a very interesting and worrying topic in small municipalities with a low population, the consequences of which deeply affect the well-being of the population, in this case the Alentejo region (Portugal).

The results support what the scientific literature has pointed out.

The article is very well documented and organized, following a correct scientific-academic structure. It also has updated and sufficient bibliographic support.

There are two small suggestions.

- In the results, divided into four dimensions, the contributions of the interviewees are very focused on the professional participants and little on the immigrants, and within these there is no representation of all the groups interviewed, only the opinions of the population from from a couple of countries, especially Ukraine and Brazil.

- It could be contrasted whether in each aspect analyzed there are differences in the opinions of the population from the other destinations interviewed.

- No conclusions appear in the work, therefore, either make some small conclusions or section 4 discussion can be called "discussion and conclusions".

This is an article that analyzes a very interesting and worrying topic in small municipalities with a low population, the consequences of which deeply affect the well-being of the population, in this case the Alentejo region (Portugal).

The results support what the scientific literature has pointed out.

The article is very well documented and organized, following a correct scientific-academic structure. It also has updated and sufficient bibliographic support.

There are two small suggestions.

- In the results, divided into four dimensions, the contributions of the interviewees are very focused on the professional participants and little on the immigrants, and within these there is no representation of all the groups interviewed, only the opinions of the population from from a couple of countries, especially Ukraine and Brazil.

- It could be contrasted whether in each aspect analyzed there are differences in the opinions of the population from the other destinations interviewed.

- No conclusions appear in the work, therefore, either make some small conclusions or section 4 discussion can be called "discussion and conclusions".

Author Response

Dear Reviewer,

Thank you again for each comment.

Regards.

Reviewer 2 Report

Comments and Suggestions for Authors

The paper’s focus is on an important policy issue which is migrant settlement and integration in rural areas so that they could be revitalized. The paper follows a qualitative methodology and comes to some interesting conclusions with policy impact but it needs an extensive and thorough revision so that it can be published in a scientific journal. Moreover, it needs extensive editing of English language so that it can be comprehensive. The paper presents some useful conclusions but there is no clear contextual basis to interpet them and highlight its contribution to the literature.

The following suggestions could be considered while revising the paper:

·        The term international migrants is more accurate than international immigrants.

·        The abstract is quite long giving much information but no indication of the theoretical background of the study.

·        The first paragraph of the introduction section refers to the research building blocks but they are derived from policy documents only rather than the state of the art literature. As a result, the contribution of the paper to the current literature is not highlighted.

·        The footnote in the first page is not accurate. International migrants pass international borders while internal migrants migrate within their country.

·        Line 43-61 could be in one paragraph in terms of context

·        The description of the approaches towards migration is very short and selective without any reference to their application in the paper. The paper mentions just a few of the several approaches developed for explaining migration flows without really engaging with them, their explanation strengths or weaknesses or their explanation value for this paper, the niche in the literature, its contribution to the literature. The paper doesn’t explain migration towards rural areas based on the referred theories.

·        Motives in section 1.2 could be explained along with the theories in section 1.1. so that the section can be more integrated and comprehensive. Moreover, migration drivers are not related to the rural areas which are the main objective of the paper.

·        There is literature about migration to rural areas (eg. S Hjort, G Malmberg, W Qi, Y Deng, B Fu, RA Bijker, T HaartsenD Strijker; T Carter, M Morrish, B Amoyaw, RA Bijker)

·        There is literature about Portugal as well (Fonseca)

·        Line 60. The term is third-country nationals.

·        Line 104. The term is the dual labor market theory.

·        The section about the attractiveness of Portugal doesn’t really contribute to the paper. A section describing drivers for migration towards rural areas with a brief mention to the case of Portugal would be more helpful to set the context for analysis.

·        An explanation of why focusing on non-EU migrants would also be helpful for the context of the study.

·        The contribution of the paper to the literature mentioned in lines 233-235 should be much more elaborated to be clear and further developed in the paper so that it can be supported.

·        Section 1.4 includes several numbers without being set in a specific context to help highlight the importance of the case of Alentejo.

·        Romanians are not EU citizens (line 199)

·        Line 215. What does revitalization population mean?

·        The importance of social work and local policy for the context which is the aim of the second research question is not explained.

·        The sample included 23 people but there were only 20 interviews? What is the value of interviewing practitioners? There is a short mention of the value of interviewing migrants (lines 260-263).

·        Line 273-274 mentions that the participants likely hold a higher education diploma. Their education was not included in the questionnaire?

·        Is the questionnaire available? Has it followed an ethics policy?

·        How the questions are related to the literature presented in the paper?

·        The sample size is an important limitation that should be mentioned in the introduction.

·        The findings are presented but not in a contextual way so that they can be explained confirming or contradicting previous literature.

·        Line 340. What is the expressive agriculture?

·        Lines 454-465 could be a different subsection

·        The description of the scope of the paper in 481-487 should also be mentioned in the introduction section along with its objectives.

·        The discussion section includes reference to literature but a restructure of the paper is necessary to facilitate the contribution of the paper to the mentioning literature.

Comments on the Quality of English Language

Extensive editing required

Author Response

(The authors gave the same response as above.)

Reviewer 3 Report

Comments and Suggestions for Authors

This paper addresses the issue of “pull factors” in international migration. Specifically, the paper investigates the reasons why international migrants (i.e., immigrants) are attracted to a rural area of Portugal, namely, the one in Alentejo, which is in the hinterland of Lisbon, the nation’s capital city. The author collects a sample of respondents, gathers qualitative data from these respondents through interviews, and analyzes these data with a well-known computer program. The results of the analysis identify the reasons why immigrants have chosen Alentejo as their destination (e.g., economic opportunity, personal security/safety) and lead to several conclusions that, the author argues, can inform policies that might lure immigrants to under-populated European regions and thereby enhance such regions’ sustainability.

While the author takes on an important and timely research topic, there are several problems that, in my opinion, seriously limit the paper’s potential to advance the literature on immigration.

The literature review (lines 27-161) does a fairly decent job of covering relevant policies and studies, yet it fails to culminate in a set of interesting hypotheses or research questions. The two research questions posed by the author (lines 230-232) do not seem very novel, and it is not clear how answering them will substantively contribute to the immigration literature. These questions might be worth answering (especially for audiences in Portugal, such as social scientists and urban planners in that country); however, the author needs to make a more compelling argument about how answers to these questions will further both theory and research on immigration. Such an argument would have to speak to the broader concerns of immigration scholars – not only to the concerns of scholars in Portugal and Europe, more generally, but also to the concerns of scholars elsewhere.

In this respect, the rationale for selecting Alentejo as the target region (lines 162-242) must be developed more fully than it is at present. I think the author is correct in arguing that immigration to Portugal has been understudied, compared to (say) immigration to larger European countries, and that there are sound reasons why immigration to Portugal should be studied (lines 162-194). Yet, these arguments are insufficient to motivate the study and fail to stipulate why Alentejo, rather than some other region in Portugal, or in Europe more generally, is the target of the investigation. In particular, I believe the author must enumerate the insights that can be gained from a study of this region that cannot be gained from a study of a rural region in (say) France, Italy or Germany.

The weakest section of the paper is the description of the data and methods (lines 243-293). The description of the sampling technique (lines 252-269) is extremely unclear and leads the reader to conclude that the author collected what amounts to a convenience “sample” of respondents rather than a true sample (i.e., one in which every member of the target population has an equal chance of being included in the study). To be sure, a non-probability sample is not a fatal flaw, especially since the goals of qualitative investigation typically do not require a representative sample (the author points this out on lines 563-565). Nonetheless, the author should provide enough information about the data collection to enable other researchers to replicate the study. The description of the method of analysis (lines 286-293) is also extremely unclear and fails to provide enough detailed information to allow the reader to understand how the interview data were analyzed.

The discussion of results (lines 294-479) is OK, but fails to yield novel insights into reasons for immigration to rural areas. Indeed, the respondents’ statements are pretty much what one would expect regarding the reasons for selecting Alentejo as their destination – for example, jobs are available, living costs are low, quality of life is high, etc. Since the results are unsurprising, the study has little potential to advance our knowledge about pull factors in immigration beyond what is already well-known. These unsurprising results, moreover, call attention to the fact that the study is profoundly limited by its non-comparative research design. If the author wants to ascertain why immigrants chose rural Alentejo versus an urban destination, then the author should have sampled immigrants in an urban destination (say, nearby Lisbon) to have a comparison group to interview and contrast with the sample of rural Alentejo immigrants. Such a comparative investigative strategy would, furthermore, be used to test alternative hypotheses about the relative attractions of rural areas for immigrants – for instance, the hypothesis that, compared to their urban counterparts, rural immigrants tend to value non-economic factors such as personal safety more highly than economic factors such as potential wages and opportunities for upward occupational mobility.

Given the above limitations, the paper’s concluding section (lines 480-568) necessarily suffers. In particular, this section is undermined because the author makes sweeping assertions that are either banal (e.g., lower living costs are attractive [line 508]) or unjustified by the data (e.g., “…for Brazilians, security is fundamental …” [line 523]). Moreover, the author makes claims about issues that were not directly addressed by the analysis, such as claims about inclusion and sustainable development (lines 540-553) and about the study’s contribution to policies and practices (lines 554-562). These claims are, at best, highly speculative since they are not solidly grounded empirically. Additionally, the author should know that simply acknowledging the study’s limits (lines 563-564) will not eliminate the inherent weakness of these limits for the investigation, and I have no idea of what allows the author to claim that “theoretical saturation was assured” (lines 564-565) by the methodology.

In spite of the aforesaid problems, I believe the author could use his/her data to perform an interesting study that compares immigrants from former Portuguese colonies in Africa with immigrants from Brazil, a former Portuguese colony in South America (see lines 175-176). I was intrigued by the finding that some Brazilians are immigrating to Portugal because of the country’s low rate of violent crime, compared to Brazil, which is becoming notorious for high rates of violent crime in its urban areas and elsewhere. The author might use the literature review (e.g., lines 162-194) to develop alternative hypotheses that are tied to the theory and research presented in the introduction (e.g., lines 68-161). Such hypotheses could be tested with a comparative research design in which African immigrants are contrasted with Brazilian immigrants with respect to the factors influencing destination choice. What I am suggesting here, of course, is a paper that is very different from the present one and not merely a revised version of the latter.

Last but not least, and with all due respect, the author must improve the writing. In its current form, the writing is atrocious, and some parts of the manuscript are almost incomprehensible. There is no way that the paper could be published in a reputable English-language journal. The author’s lack of attention to detail is also evident in numerous misspellings and the inconsistency of the bibliography’s format (lines 571-701) with the citation style used in the manuscript’s text.

Comments on the Quality of English Language

Very poor -- nearly incomprehensible in some parts of the manuscript.

Author Response

(The authors gave the same response as above.)

Reviewer 4 Report

Comments and Suggestions for Authors

This article looks at the motives for migration to rural regions of Portugal.  The article is well situated in the academic literature, identifying its contribution to the academic discourse on migration and the gap in the literatre it hopes to fill.  The methods are appropriate for the research question.  I worry that the sample size is small, but I think the conclusions are consistent with the academic literature, which limits the concerns over a small sample size.  Again, the authors do a fine job of situating their results in the academic literature and identify where they contribute to discussions of migration.  

I have two minor concerns with the paper as written.  First, I would recommend the authors proofread the paper with an eye towards redundancy.  There are many points in the paper where the authors restate ideas seen in the preceding paragraph.  I think this would help with readability.  

My second concern is with the study area.  I think the readers would benefit from a brief discussion of the Alentejo.  What makes the region attractive to migrants?  Where is the region in Portugal?  Also, I think a map would help the readers understand the context much better.  

Overall, I think the paper is well researched and well written.  The authors provide a fine literature review and situate their research in the academic literature.  Thank you for the opportunity to review your paper.  Best of luck in your future projects.  

Comments on the Quality of English Language

Overall, the paper is well written.  I had some concerns with redundancy throughout the paper.  I think a proofread of the paper will help identify the sections and tighten up the final draft.  

Author Response

(The authors gave the same response as above.)

Round 2

Reviewer 2 Report

Comments and Suggestions for Authors

Although the authors made a decent effort to revise the manuscript, it still lacks academic soundness and logical coherence particularly due to its weakness in terms of critical engagement with migration literature . Main flaws of the paper haven’t been adequately addressed especially with regard to the formation of the research hypotheses and its contribution to migration scholarship.

Comments on the Quality of English Language

English language editing is necessary

Author Response

Dear Reviewer 2,
The manuscript was revised accordingly and a proofreading service was provided again.
In fact, your comments during this process supported a substantial improvement of the manuscript. Thank you for the time spent and your dedication. 

Reviewer 3 Report

Comments and Suggestions for Authors

I'm not sure that all my concerns are fully addressed in this revised version, but it does appear that the authors have worked hard to improve the paper.

I recommend that the paper be accepted, subject to major copy-editing to improve the writing.

  Comments on the Quality of English Language

Still not very good.

Author Response

  Dear Reviewer 3,
The manuscript was revised and a proofreading service was provided again.
In fact, your comments during this process supported a substantial improvement of the manuscript. Thank you for the time spent and your dedication. 

Round 3

Reviewer 2 Report

Comments and Suggestions for Authors

The authors have managed to improve the manuscript but it still lacks scientific soundness.

The authors could consider the following suggestions to further develop the paper so that it can be published in the journal:

·        As mentioned in previous rounds of review, there seems to be a vague selection of migration theories to support the findings of the paper instead of following a specific solid theoretical background to formulate the research questions of the interviews and then interpreting the answers to conclude whether they confirm the theory or contradict it. The authors have followed a pattern of analysis as developed by the review article of Wadood et al. (2021) they cite, without referring to the original papers developing each approach or to the whole set of theories in each level of analysis.

·        Moreover, I am not sure whether the level of analysis followed serves the purpose of this specific paper.

·        As regards the drivers of migration, has the classification followed a pattern suggested in the literature or the authors have classified the drivers as reported in the literature in six dimensions?

·        Are the six dimensions related to the theories the authors have chosen?

·        How the theories and the motives are related to the research questions?

·        A useful classification of the drivers of migration could be found here: https://www.migrationdataportal.org/themes/migration-drivers

·        The formulation of the first hypothesis is rather general. Since the sample is so small and focused in a specific region and if Portugal’s attractiveness is taken as a fact, then I wouldn’t expect the results to be generalized to explain migration to other rural areas. Moreover, there are only a few migration approaches presented in the paper, so the authors intend to understand whether these specific ones explain migration to Alentejo?

·        Although Lines 178-184 seem to support the second hypothesis, they are not included in the discussion of section 1.2 but they seem rather as a supplement. Since Alentejo has also commerce, industry, construction sectors why choosing this one in the hypothesis?

·        Is the case of Portugal as an attractive place to migrate considered as a base upon which the rest of the factors attracting migrants are examined or a question to examine? How is the third hypothesis related to this? How the authors interpret the results as regards the attractiveness of the country based on the theory presented in the paper?

·        Since this paper’s contribution is considered to be its policy impact, I would suggest the significance of migration for rural areas to be underlined adequately. In lines 60-65 the positive impact is briefly mentioned and in lines 80-82 the negative consequences are briefly stressed. The methodological process doesn’t include that aspect. So the findings of the literature that support this argument could be discussed extensively in a section along with the lines 298-303.

·        Is the education information of the practitioners important for the context?

·        The fact that the migrant participants work on academia, accounting, factories, hairdressing, social services and security doesn’t reject the second hypothesis?

·        The dimensions of the motives describing the results of the interviews are different than those presented in section 1.2?

Round 3

Reviewer 2

The authors have managed to improve the manuscript but it still lacks scientific soundness.

 The authors could consider the following suggestions to further develop the paper so that it can be published in the journal: Thank you for the relevant suggestions to improve our article. We conducted a significant revision.

  • As mentioned in previous rounds of review, there seems to be a vague selection of migration theories to support the findings of the paper instead of following a specific solid theoretical background to formulate the research questions of the interviews and then interpreting the answers to conclude whether they confirm the theory or contradict it. The authors have followed a pattern of analysis as developed by the review article of Wadood et al. (2021) they cite, without referring to the original papers developing each approach or to the whole set of theories in each level of analysis. We revised the manuscript according to the comment.
  • Moreover, I am not sure whether the level of analysis followed serves the purpose of this specific paper.
  • As regards the drivers of migration, has the classification followed a pattern suggested in the literature or the authors have classified the drivers as reported in the literature in six dimensions? Are the six dimensions related to the theories the authors have chosen? We revised the manuscript according to the comment.
  • How the theories and the motives are related to the research questions? We revised the text.
  • A useful classification of the drivers of migration could be found here: https://www.migrationdataportal.org/themes/migration-drivers

Thank you very much for sharing.

  • The formulation of the first hypothesis is rather general. Since the sample is so small and focused in a specific region and if Portugal’s attractiveness is taken as a fact, then I wouldn’t expect the results to be generalized to explain migration to other rural areas. Moreover, there are only a few migration approaches presented in the paper, so the authors intend to understand whether these specific ones explain migration to Alentejo? The study design did not intent neither to generalise its results nor representativeness, as presented in the manuscript. We conducted a brief review and did a selection of diverse theoretical approaches to understand motives to ones move to another place/ country. There are other approaches however there is no room for all of them regarding the aim of this study. We revised the text according to the comment, thank you.
  • Although Lines 178-184 seem to support the second hypothesis, they are not included in the discussion of section 1.2 but they seem rather as a supplement. Since Alentejo has also commerce, industry, construction sectors why choosing this one in the hypothesis? The hypothesis seems relevant since economic reasons to migrate are consensual in the literature. And specifically, agriculture, and its industry, is consistently addressed by the literature on migration in rural areas as a main reason to explain the presence of immigrants. Thus, the authors intent relevant to explore in the study. Section 1.2. approaches economic and employment motives influencing migration, in general terms. At its end, and specifically stressing rural areas, we present motives influencing migration to rural areas. Moreover, we revised the text according to the comment.
  • Is the case of Portugal as an attractive place to migrate considered as a base upon which the rest of the factors attracting migrants are examined or a question to examine? How is the third hypothesis related to this? How the authors interpret the results as regards the attractiveness of the country based on the theory presented in the paper? We revised the text according to the comment.
  • Since this paper’s contribution is considered to be its policy impact, I would suggest the significance of migration for rural areas to be underlined adequately. In lines 60-65 the positive impact is briefly mentioned and in lines 80-82 the negative consequences are briefly stressed. The methodological process doesn’t include that aspect. So the findings of the literature that support this argument could be discussed extensively in a section along with the lines 298-303. We revised it according to the comment, thank you.
  • Is the education information of the practitioners important for the context? We removed it.
  • The fact that the migrant participants work on academia, accounting, factories, hairdressing, social services and security doesn’t reject the second hypothesis? The second hypothesis is based on literature and suggests that the motives for a non-EU citizen to go to rural areas extend beyond agricultural employment and economic reasons, so it is confirmed by the empirical results of the study, since, several other dimensions of factors influencing immigrants’ emerged.
  • The dimensions of the motives describing the results of the interviews are different than those presented in section 1.2? Some are aligned (e.g., employment opportunities) and others differ (e.g., support from social workers). Section 1.2 are based on migration in general and the results present aspects focused mainly on rural regions as the Alentejo. The discussion section present conclusions bases on the analysis. Moreover, we revised the text according to the comment.

Best regards.